# Effects of the Entomopathogenic Fungus *Mucor hiemalis* BO-1 on the Physical Functions and Transcriptional Signatures of *Bradysia odoriphaga* Larvae

**DOI:** 10.3390/insects14020162

**Published:** 2023-02-08

**Authors:** Guodong Zhu, Wenjuan Ding, Haipeng Zhao, Ming Xue, Pengfei Chu, Liwei Jiang

**Affiliations:** 1College of Agronomy, Liaocheng University, Liaocheng 252000, China; 2College of Plant Protection, Shandong Agricultural University, Tai’an 271018, China

**Keywords:** entomopathogenic fungi, *Bradysia odoriphaga*, *Mucor hiemalis* BO-1, feeding capacity, antioxidant enzymes, transcriptome analysis

## Abstract

**Simple Summary:**

*Mucor hiemalis* BO-1 was acutely toxic to *Bradysia odoriphaga* larvae and was as effective as some chemical pesticides. *Mucor hiemalis* BO-1 infection inhibited the feeding capacity of *Bradysia odoriphaga* larvae, reduced food consumption and nutrient contents, and caused a decline in digestive enzyme activity. Transcriptome analysis of diseased *B. odoriphaga* larvae indicated significant changes in the expression of the genes related to the digestive process, energy metabolism, antioxidant response, and immune processes.

**Abstract:**

*Mucor hiemalis* BO-1 is an entomopathogenic fungus that infects *Bradysia odoriphaga*, a destructive root maggot. *M. hiemalis* BO-1 possesses stronger pathogenicity to the larvae than to other stages of *B. odoriphaga*, and provides satisfactory field control. However, the physiological response of *B. odoriphaga* larvae to infection and the infection mechanism of *M. hiemalis* are unknown. We detected some physiological indicators of diseased *B. odoriphaga* larvae infected by *M. hiemalis* BO-1. These included changes in consumption, nutrient contents, and digestive and antioxidant enzymes. We performed transcriptome analysis of diseased *B. odoriphaga* larvae, and found that *M. hiemalis* BO-1 showed acute toxicity to *B. odoriphaga* larvae and was as toxic as some chemical pesticides. The food consumption of diseased *B. odoriphaga* after inoculation with *M. hiemalis* spores decreased significantly, and there was a significant decrease in total protein, lipid, and carbohydrates in diseased larvae. Key digestive enzymes (protease, α-amylase, lipase, and cellulase) were significantly inhibited during infection. Peroxidase maintained high activity, and the activity of other antioxidant enzymes (catalase, superoxide dismutase, and glutathione S-transferases) first increased and then decreased. Combined with the transcriptional signatures of diseased *B. odoriphaga* larvae, *M. hiemalis* BO-1 infection resulted in decreased food consumption, reduced digestive enzyme activity, and altered energy metabolism and material accumulation. Infection was also accompanied by fluctuations in immune function, such as cytochrome P450 and the Toll pathway. Therefore, our results laid a basis for the further study of the interactions between *M. hiemalis* BO-1 and *B. odoriphaga* and promoted the genetic improvement of entomopathogenic fungi.

## 1. Introduction

Biological control is an important part of integrated pest management (IPM). Successful biological control helps protect the environment and advances harmonious interaction between man and nature [1,2]. Entomopathogenic fungi have been used in biological control programs against several agricultural pests. Some fungal biocontrol agents, such as *Beauveria bassiana* and *Metarhizium anisopliae*, have replaced chemical insecticides for the control of agricultural pests [3,4,5]. However, the limitations of entomopathogenic fungi, such as high environment dependence and protracted insecticidal activity, have slowed their development and use. Due to this, only a few fungal species have been successfully developed as biopesticides. Many entomopathogenic fungi possess good pathogenicity in the laboratory, but their field efficacy depends on favorable environmental conditions, such as temperature, humidity, and characteristics of the microorganism [6,7,8]. For example, the optimal temperatures of *Beauveria bassiana* that provide the best control efficiency range from 25 to 30 °C [9,10]. Hence, it is necessary to find an entomopathogenic fungus with high pathogenicity and broad environmental adaptability.

Most entomopathogenic fungi infect host insects directly by penetration through the cuticle into the hemocoel. Some pathogens can infect host insects via the oral route during food ingestion [11,12]. Once the pathogens invade the hemocoel, they switch to yeast-like hyphal bodies to overcome, evade, or suppress insect immune defenses, and then grow to form mycelia. This ultimately kills the host, and sporulation occurs in the cadaver [3,13]. Generally, in the course of the disease, entomopathogenic fungi can proliferate rapidly by robbing the nutrients of the host and secreting mycotoxins. The mycotoxins damage the free amino acids in the hemolymph, disturb the storage and utilization of nutrients, and interfere with important metabolic enzymes [14,15]. Insects rely on the combined efforts of physical barriers (i.e., cuticle surface and chitin barriers of the gut) and both cellular and humoral components of their innate immune system to protect themselves from invasion and death due to microorganisms [16]. However, the immune response of insects involves high consumption of energy and inhibits the normal life activities of host insects [17,18]. *B. bassiana* infection elicits immune responses from both the cellular and humoral arms of the insect immune system, including antioxidant enzymes, phenoloxidase (PO) cascade, and cytochrome P450 [19,20]. The ability to infect through both cuticular and oral routes would make fungal pathogens a more difficult challenge to the insects’ immune system and thereby a good candidate for a biopesticide. Entomopathogenic fungi infection can also cause dietary stress since conidia represent indigestible material for larvae. They occupy the digestive tract and limit host access to ingested nutrients [5,11]. Consideration of the mechanisms of pathogenesis is important for the determination of fungal virulence factors that might be manipulated to increase the speed of mycosis and death to levels similar to those produced by chemical insecticides.

*Bradysia odoriphaga* Yang et Zhang is a root maggot that causes serious losses to vegetables, flowers, and edible fungi [21,22]. The larvae can directly damage plants by feeding on the root and corm tissues and causing wilt or rot. Uncontrolled Bradysia infestation can cause more than 50% yield reduction in vegetables and edible fungi [23]. Since the damage caused by larvae is primarily on the underground parts of plants, they are difficult to control effectively [22,24]. Chemical insecticides remain the most important component in *B. odoriphaga* larvae control programs. The chemical products include organophosphates (phoxim and chlorpyrifos), carbamates (carbosulfan), and neonicotinoids [25], despite direct and indirect toxic effects on nontarget organisms and humans. However, after many years of application, larval pesticide resistance has developed [21]. In addition, conventional chemical pesticide use is now restricted due to environmental pollution and human health concerns. Therefore, there is a need for alternative biocontrol approaches.

Compared to other agricultural pests, there are few effective biological control agents of *B. odoriphaga*. The list includes *Beauveria bassiana*, *Bacillus thuringiensis*, *Stratiolaelaps scimitus* (predatory mite) and entomopathogenic nematodes [21,26]. Only *B. bassiana*, *B. thuringiensis*, and entomopathogenic nematodes have been used for field control of *B. odoriphaga* [21]. The field control effects of these biocontrol agents are often unsatisfactory due to poor efficacy and slow action. Zhou et al. (2014) reported that when the dosage of *B. bassiana* granules (15 billion spores/g) was 4.5 kg/ha, the field control efficiency was 81.15% (7 d after application) and 85.19% (21 d after application) [27]. The field control efficiency of three species of entomopathogenic nematodes (*Heterorhabditis bacteriophora, Steinernema carpocapsae*, and *Steinernema feltiae*) to *B. odoriphaga* was 40–50% at 14 d after application [26]. The discovery and development of a new biological control agent possessing high pathogenicity and strong environmental adaptability against *B. odoriphaga* will be necessary to improve the control of *B. odoriphaga*.

*Mucor hiemalis* BO-1 is a recently discovered entomopathogenic fungus that possesses strong pathogenicity to *B. odoriphaga* larvae [28]. Field tests confirmed that *M. hiemalis* BO-1 produced excellent control of *B. odoriphaga* and had great potential for further evaluation as a biocontrol agent. *M. hiemalis* was also reported to be pathogenic against other arthropods, such as *Latrodectus geometricus* Koch [29], *Evergestis extimalis* Scopoli [30], and Nemouroidea [31]. After larval inoculation with *M. hiemalis* spores, *B. odoriphaga* movement slowed, and the body became transparent. During infection, food residue in the alimentary canal decreased [28]. It was hypothesized that *M. hiemalis* BO-1 infection disrupted the physiological and digestive functions of *B. odoriphaga* larvae. However, little is known about the physiological responses of *B. odoriphaga* larvae to pathogen infection and the molecular mechanisms underlying the pathological processes. This lack of information limited the further development of this strain.

In this study, we evaluated the pathogenicity of *M. hiemalis* BO-1 to fourth-instar larvae of *B. odoriphaga*. Physiological indicators of diseased *B. odoriphaga* larvae, such as food consumption, nutrient contents, digestive enzymes, and antioxidant enzymes, were studied to help understand the effects of *M. hiemalis* BO-1 infection on the physiological functions of *B. odoriphaga* larvae. The transcriptional signatures of diseased *B. odoriphaga* larvae were analyzed, and a detailed bioinformatics analysis was conducted on partially important differentially expressed genes (DEGs).

## 2. Materials and Methods

### 2.1. Insect Materials

The samples of *B. odoriphaga* were originally collected from a Chinese chive field in Liaocheng, Shandong, in April 2019. The samples were reared on Chinese chives for more than 20 generations. Eggs, larvae, and pupae were reared in culture dishes (Φ = 9 cm), and newly emerged adults were placed in pairs in oviposition containers (3 cm diameter × 1.5 cm high). Insect samples were maintained in growth cabinets at 23 ± 1 °C with 75 ± 5% relative humidity.

### 2.2. Entomopathogenic Fungi Strain

*Mucor hiemalis* BO-1 strain was isolated from infested *B. odoriphaga* larvae and deposited at China General Microbiological Culture Collection Center (CGMCC, No. 40036). This strain was cultivated on potato dextrose agar (PDA) plate (Φ = 9 cm) in Petri dishes at an optimal temperature of 23 °C. After 10 d, 50 mL 0.1% Tween 80 distilled water was added to the dish, and the surface of the colony was gently and repeatedly scraped with a Petri scalpel. The suspension was filtered using three-layer filter paper, and the filtrate was gathered as a spore suspension. The spore suspension concentration was calculated through blood counting chamber analysis.

### 2.3. Pathogenicity Assays

Serial spore suspension dilutions (1 × 10^3^, 1 × 10^4^, 1 × 10^5^, 1 × 10^6^, 1 × 10^7^, and 1 × 10^8^ spores/mL) were made with 0.1% Tween 80 distilled water. Bioassays on 4th-instar larvae were conducted using standard contact and stomach bioassay methods [23]. One piece of filter paper (Φ = 9 cm) was moisturized by dropping 1 mL spore suspension and placed in the culture dish. Fresh diet was placed on the filter paper. Larvae were placed around the fresh diets. Pure water treatment was used as the control. Each treatment contained 5 replicates with 25 individuals. The survival number of *B. odoriphaga* larvae was recorded daily. The median lethal time (LT_50_) and median lethal concentration (LC_50_) values were calculated according to probit analysis using SPSS Statistics 18.0.0 (2009; SPSS Inc., Quarry Bay, Hong Kong, China).

### 2.4. Effects of M. hiemalis BO-1 on Food Consumption

According to the pathogenicity results, the survival rates of fourth-instar larvae at 24, 48, 72, and 96 h after inoculation were 98.75%, 90%, 78.75%, and 63.75% for 10^4^ spores/mL treatment and 97.5%, 86.25%, 68.75%, and 50% for 10^5^ spores/mL treatment, respectively. At 96 h, the survival rate of infected larvae was close to 50%, and so the two spore concentrations of 10^4^ spores/mL and 10^5^ spores/mL were used for the evaluation tests on the treated larvae. Newly emerged 4th-instar larvae were exposed to an *M. hiemalis* BO-1 spore suspension (1 × 10^4^ and 1 × 10^5^ spores/mL). The food consumption of larvae was measured using the method described by Zhao et al. (2018) [23] with slight modifications. A total of 50 larvae were transferred to fresh Petri dishes, and fresh Chinese chive rhizomes were weighed and used to feed the larvae. The diets were changed every 24 h, and the remaining rhizomes were reweighed. The food consumption of every larva was calculated according to the variation in Chinese chive rhizome weight/the larva number. Every treatment contained five replicates. The 0.1% Tween 80 treatment was used as the control.

### 2.5. Biochemical Assay of Nutrient Contents in Larvae

Sample preparation: The larvae were treated with 1 × 10^4^ and 1 × 10^5^ spores/mL of *M. hiemalis* BO-1 spore suspension. After 24, 48, 72, and 96 h, groups of 100 surviving larvae were weighed and used for the analysis of biochemical indicators. Every treatment contained three replicates.

Lipid amount: The 100 larvae were homogenized and extracted with 1.5 mL chloroform–methanol (2:1) solution. After centrifugation at 5000 rpm for 10 min, the supernatant was dried at 40 °C. The residuum was dissolved with 1.0 mL H_2_SO_4_ (98%) and incubated at 90 °C for 10 min. Then, 50 μL of each sample was mixed with 250 μL of vanillin, and the absorbance was read at 530 nm.

Carbohydrate amount: The 100 larvae were homogenized in cold 800 μL of trichloroacetic acid (10%), followed by centrifugation at 8000 rpm for 10 min. Then, 100 μL of the supernatant and 210 μL of 10% trichloroacetic acid were mixed, and 1.8 mL of 0.2% anthrone was added to the mixture. The mixture was placed in a boiling water bath for 10 min. The absorbance was read at 630 nm. A standard curve was established with glucose.

Protein content: The 100 larvae were homogenized in 0.05 M phosphate buffer. The crude homogenates were centrifuged at 5000× *g* for 15 min. The protein concentration of the supernatant was determined by the Bradford assay [32].

### 2.6. Enzyme Activity Assay

Sample preparation: Larvae were treated by the method described in Section 2.4. After 12, 24, 36, 48, 72, and 96 h, the surviving larvae were dissected on ice, and the entire digestive tracts of 50 larvae were collected as the sample for digestive enzymes assay. At each test interval, 30 surviving larvae were collected as the sample for the antioxidant enzyme activity assay. All the samples were homogenized in 0.05 M cold phosphate buffer containing 1% polyvinylpyrrolidone (pH 7.8, 0–4 °C). The crude homogenates were centrifuged at 10,000× *g* for 10 min at 4 °C. The supernatants were collected, and these represented the enzyme samples. Each treatment had five replicates. Protein concentrations of each sample were determined using the Bradford assay [32].

Digestive enzyme activity assay: The digestive enzyme activities were detected using the α-Amylase Assay Kit, Lipase Assay Kit, Protease Assay Kit, and Cellulase (CL) Assay Kit (Nanjing Jiancheng Bioengineering Institute, Nanjing, China). For the protease activity assay, casein was used as the substrate, and tyrosine was detected as the product of enzymatic reactions. The amount (nmol) of tyrosine produced per mg protein catalyzing casein decomposition per min at 660 nm and 37 °C was defined as the protease activity. For the α-amylase activity assay, soluble starch was used as the substrate, and maltose was detected as the product of enzymatic reactions. The α-amylase activity was defined as the amount of maltose produced by soluble starch decomposition catalyzed by per mg protein per min at 540 nm and 37 °C. For lipase activity assay, 4-nitrophenyl butyrate was used as the substrate, and p-nitrophenol was detected as the product of enzymatic reactions. The amount of p-nitrophenol that was produced by the enzymatic reaction catalyzed by per mg protein per min at 405 nm and 37 °C was defined as the lipase activity. For the cellulase activity assay, cellulose was used as the substrate, and reducing sugar (glucose) was detected as the product by the color reaction at 540 nm. The amount of glucose that was produced by the enzymatic reaction catalyzed by per mg protein per min at 540 nm and 37 °C was defined as the cellulase activity.

Antioxidant enzyme activity assay: The antioxidant enzyme activities were detected using the Catalase (CAT) Assay Kit, Peroxidase (POD) Assay Kit, Superoxide Dismutase (SOD) Assay Kit, and Glutathione S-Transferase (GST) Assay Kit (Solarbio Science Technology Company, Beijing, China). The operation sequences were performed following the kit instructions. The amount (μmol) of H_2_O_2_ decomposition at 240 nm per min per mg protein was defined as one unit of CAT activity. One unit of POD activity was defined as the amount that catalyzes 1 μmol of guaiacol oxidation at 470 nm per minute per mg protein. One unit of SOD activity was defined as the amount of enzyme that caused 50% inhibition of the nitro blue tetrazolium reduction at 550 nm. GST activity was tested using 1-chloro-2,4-dinitrobenzene (CDNB) as substrate. The reaction mixture contained 75 μL of 0.6 mM CDNB, 150 μL of 6 mM reduced glutathione (GSH), and 50 μL of enzyme solution. The OD value was monitored at 340 nm and 27 °C for 5 min.

### 2.7. Transcriptome Analysis

Sample preparation: Fourth-instar *B. odoriphaga* larvae were infested with an *M. hiemalis* BO-1 spore suspension (1 × 10^5^ spores/mL). After 48 h, 100 surviving larvae were collected as the sample. The treatment without spores was used as the control. Each treatment contained two replicates. All of the samples were immediately immersed in liquid nitrogen. Total RNA was extracted using the Total RNA Isolation TRIzol Kit (Thermo Fisher Scientific Biotech, Shanghai, China). The RNA quality was monitored on 1% agarose gel, and RNA purity was confirmed using a NanoPhotometer spectrophotometer (Appendix A). First-strand cDNA was synthesized from mRNA after enrichment by fragmentation with buffer and oligo (dT) magnetic beads. The double-stranded cDNA was purified, which was then eluted with EB buffer for end-repair and addition of poly (A). The library was sequenced on an Illumina HiSeqTM 3000 (Illumina, Beijing Novogene Biotech Co., Ltd. (Beijing, China). The raw data were deposited in the NCBI Short Read Archive (SRA) database with accession number: PRJNA922622.

Transcriptome analysis: To clean up the mRNA derived from the fungus or other microbiota in larvae, clean reads were mapped to the *B. odoriphaga* genome (Bioproject ID: PRJNA612767) using Hisat 2.0.5 software designed by the Center for Computational Biology of Johns Hopkin University (WA, USA) [33]. The mapped reads were normalized using the DESeq 1.18.0 package. Gene-expression levels were calculated as fragments per kilobase per million (FPKM) using cufflinks 2.1.1 software. Differentially expressed genes (DEGs) were determined based on their expression abundance in different treatment groups. The difference in gene expression between the treatment groups was deemed significant if the *p*-value < 0.05. Hierarchical clustering analysis of DEGs was performed to cluster genes that exhibited the same or similar expression levels. Gene Ontology (GO) was applied to analyze the function of DEGs based on the GO database and to determine the biological implication of genes in the significant or representative profiles. The identification of pathways for DEGs was according to the Kyoto Encyclopedia of Genes and Genomes (KEGG) database. The resulting *p*-values were adjusted using the false discovery rate (FDR) algorithm. Then, pathway categories with *p*-values < 0.01 and FDRs < 0.05 were reported.

Quantitative RT-PCR analysis: To validate the results of the transcriptome analysis, 12 DEGs were analyzed using qRT-PCR. New biological samples were prepared according to the methods described in Section 2.4. Total RNA isolation and cDNA synthesis were performed, and the fluorescence quantitative PCR primer sequences of these genes are listed in Appendix A. Real-time PCR was performed using the One Step SYBR^®^ Prime Script TMPT-PCR kit (Takara, Dalian, China). The housekeeping gene β-actin was used as the internal control. Transcript levels were determined in relation to that of β-actin using the 2^−△△Ct^ method.

### 2.8. Statistical Analysis

We tested the other variables for homogeneity of group variances using Levene’s test and normality using the Kolmogorov–Smirnov test before statistical analyses. The significant difference analysis was performed with SPSS Statistics 18.0.0 (2009; SPSS Inc., Quarry Bay, Hong Kong, China).

## 3. Results

### 3.1. Pathogenicity of M. hiemalis BO-1

*M. hiemalis* BO-1 was highly pathogenic to *B. odoriphaga* larvae (Table 1). The larvae reached a high mortality rate in a short time, even at a low spore inoculum concentration. At observation times 3 d and 5 d, the median lethal concentrations (LC_50_) were 2.180 × 10^6^ and 0.965 × 10^5^ spores/mL, respectively. When the spore concentrations were 1 × 10^4^, 1 × 10^6^, and 1 × 10^8^ spores/mL, the median lethal times (LT_50_) were 4.79, 2.99, and 2.43 d, respectively.

### 3.2. Effects of M. hiemalis Infection on Food Consumption and Nutrient Contents of B. odoriphaga

Food consumption: The food consumption of *B. odoriphaga* larvae was significantly decreased at 48 h after inoculation with *M. hiemalis* spores (Figure 1A). High spore-concentration treatment (1.0 × 10^5^ spores/mL) inhibited food consumption more than the low-concentration treatment (1.0 × 10^4^ spores/mL). At 72 h after the initial inoculation, the food consumption decreased by 48.19% (1.0 × 10^4^ spores/mL treatment) and 66.97% (1.0 × 10^5^ spores/mL treatment). However, at 120 h, the food consumption of the 1.0 × 10^5^ spores/mL treatment group was higher than that of the 1.0 × 10^4^ spores/mL treatment group.

Nutrient contents: *M. hiemalis* infection inhibited nutrient accumulation in the larval body or accelerated nutrient consumption (Figure 1B–D). During normal larval development, proteins, lipids, and carbohydrates gradually accumulate. After *M. hiemalis* infection, the total protein amount stopped increasing, although the larvae continued feeding (Figure 1B). At 48 h after inoculation, the lipid amounts in the treated groups reached a maximum, and there were no significant differences between the control and treatment groups. The lipid amount of the two inoculation groups decreased rapidly, and significant differences were discovered at 72 and 96 h compared to the control group (Figure 1C). The carbohydrate amount of the two inoculation groups gradually decreased after infection and was lower than the control group at 72 and 96 h (Figure 1D). Significant dosage effects showed that the higher the spore concentration, the more obvious the decrease in the three nutrient contents.

### 3.3. Effects of M. hiemalis Infection on Digestive Enzyme Activities

The effects of *M. hiemalis* infection on the digestive enzyme activities of *B. odoriphaga* larvae are shown in Figure 2. The activities of protease, lipase, α-amylase, and cellulase significantly decreased with increased infection time. After 36 h, all of the digestive enzyme activities of the inoculation groups had decreased significantly compared to the control group. However, at 24 h, the decreases in lipase, α-amylase, and cellulase enzymatic activities were less obvious than the decrease in protease activity. At 72 h after inoculation, the protease activities of the two inoculation groups decreased by 32.91% (10^4^ spores/mL) and 57.34% (10^5^ spores/mL) compared to the control group. The α-amylase activities decreased by 72.58% and 85.12%, cellulase activities decreased by 63.89% and 78.40%, and lipase activities decreased by 41.09% and 69.24%. Significant dosage effects were also observed. The higher the spore concentration, the more obvious the decrease in the four digestive enzyme activities.

### 3.4. Effects of M. hiemalis Infection on Antioxidant Enzyme Activities

The activities of CAT, POD, SOD, and GST in *B. odoriphaga* 4th-instar larvae were determined after infection with *M. hiemalis* spores (Figure 3). POD activity increased after infection; the activity increased by 42.51% (1.0 × 10^4^ spores/mL) and 66.47% (1.0 × 10^5^ spores/mL) compared to the control group at 36 h, and 19.83% (1.0 × 10^4^ spores/mL) and 41.90% (1.0 × 10^5^ spores/mL) at 72 h. However, CAT, SOD, and GST activities fluctuated after infection. Their trends were similar, with an initial increase followed by a decline. The CAT activity increased by 84.22% (1.0 × 10^5^ spores/mL) and 33.57% (1.0 × 10^4^ spores/mL) at 24 h after infection and decreased by 47.78% (1.0 × 10^5^ spores/mL) and 36.33% (1.0 × 10^4^ spores/mL) at 72 h after infection. The SOD and GST activity of the 1.0 × 10^5^ spores/mL treatment group increased by 27.88% (SOD) and 36.04% (GST) at 24 h after infection but decreased by 45.08% (SOD) and 51.66% (GST) at 72 h.

### 3.5. Transcriptome Analysis

Quality assessment of the transcriptome: The control 2 library produced the most data (51,677,542 clean reads), while the treatment 2 library produced the fewest clean reads (44,925,166). All libraries exhibited good quality, with 94.19% to 94.33% of the clean reads meeting base call quality at the Q30 standard. According to the alignments of sequences in the unigene library, the statistics of mapped reads in each sample are shown in Table 2. The mapped ratio was 87.89 to 89.39%.

Identification and annotation analysis of DEGs: Based on the DEG analysis, 377 DEGs between *M. hiemalis* infection and control treatment were identified (Appendix A). Of these, 214 unigenes were upregulated, and 163 unigenes were downregulated. For GO analysis, we annotated DEGs into three GO categories: biological process (BP), cellular component (CC), and molecular function, which were divided into 44 subgroups (Figure 4A). For the BP category, the two main enriched terms were the cellular process and metabolic process, and the number of DEGs was 81 and 82. For the CC category, the main enriched terms were the cell and cell part, and the number of DEGs in both terms was 95. For the molecular function (MF) category, the main enriched terms were catalytic activity and binding, and the number of DEGs was 106 and 72. The biochemical pathways of the DEGs were investigated using the KEGG database. The top 10 enriched KEGG pathways were the PPAR signaling pathway, metabolic pathways, lysosome, fatty acid metabolism, phagosome, ferroptosis, glycerolipid metabolism, fatty acid degradation, metabolism by cytochrome P450, and steroid biosynthesis (Figure 4B, Appendix A). In the metabolic pathways, 23 DEGs were identified (Appendix A). Genes including lipid catabolic processes (HA402-011554, HA402-013349), carbohydrate metabolic processes (HA402-007273, HA402-007340, HA402-005954), glutathione metabolic processes (HA402-015641, HA402-005674), glucuronosyltransferase activity (HA402-015641, HA402-015641), or phospholipid metabolic processes (HA402-011554) were partially repressed. Genes including trehalose metabolic processes (HA402-003401, HA402-006219, HA402-013707), monosaccharide metabolic processes (HA402-009149), lipid biosynthetic processes (HA402-002025), dopamine metabolic processes (HA402-005954, HA402-001421), or doxorubicin metabolic processes (HA402-001272) were upregulated.

Reverse transcription (RT-PCR) validation: To further validate the gene expression profiles, 12 candidate differentially regulated genes (listed in Table 3) with different biological functions were analyzed using RT-PCR. The DEGs (*amyA*, *PRSS*, and *LIPA*) related to the digestion and carbohydrate metabolic process were down-regulated in *B. odoriphaga* larvae. This indicated that *M. hiemalis* infection decreased the insects’ ability to digest nutrients (lipids, starch, and protein). The relative expression of *treA* and *CPT1A* related to the metabolic process were up-regulated by 1.72- and 1.29-fold compared to the control group, respectively. The DEGs (*PRDX5*, *catB*, and *gst*) related to the response to the oxidative stress process were up-regulated, while *SOD1* was down-regulated indicating high antioxidant capacity (Figure 5). The DEGs (*CYP9* and *ECSIT*) related to drug metabolic and innate immune responses were up-regulated by 2.24- and 1.37-fold indicating significant immune responses caused by *M. hiemalis* infection. These RT-PCR results were similar to the RNA-seq data (detail shown in Table 3).

## 4. Discussion

Entomopathogenic fungi represent a novel way to control the root maggot *B. odoriphaga*. *M. hiemalis* BO-1 is a newly identified entomopathogenic fungus with high pathogenicity to *B. odoriphaga*. This fungus has good potential for commercial development and practical use [28]. Our bioassay results revealed that *M. hiemalis* BO-1 produced high mortality in *B. odoriphaga* larvae. When the spore concentration was 1.0 × 10^5^ to 1.0 × 10^6^ spores/mL, the median lethal time (LT_50_) was 2.99 to 3.79 d (Table 1). This was comparable to some neonicotinoid and organophosphorus pesticides [21,22]. A 3 d post-treatment interval is often used to evaluate the activity of chemical insecticides against *B. odoriphaga* [21,23]. The test time of most biological and “reduced-risk” insecticides is five days or more, and this includes insect growth regulators and botanical insecticides [22,34]. However, the toxicity of known entomopathogens to *B. odoriphaga* is lower than chemical insecticides. Zhou et al. (2014) reported that the LT_50_ of *B. bassiana* (1.0 × 10^7^ spores/mL) against *B. odoriphaga* larvae was 14.51 d [27]. The test time used to evaluate the infection ability of entomopathogenic nematodes to *B. odoriphaga* larvae was 10 d [35]. To improve the efficacy of control agents, some studies have applied a mixture of chemicals and biocontrol agents [26].

During infection, food residue in the alimentary canal of *B. odoriphaga* larvae decreased, and the larval body became transparent. *M. hiemalis* BO-1 infection disrupted the physiological and digestive functions of *B. odoriphaga* [28]. In the present study, larval food consumption decreased significantly during *M. hiemalis* BO-1 infection, and the nutrient substances (proteins, lipids, and sugars) were reduced (Figure 1). Previous studies reported that *Beauveria* spp. and *Metarhizium* spp. infection reduced feeding of the host insects. These species included *Anopheles gambiae* [36], *Locusta migratoria* [5], *Dichroplus maculipennis* [4], and *Spodoptera littoralis* [37]. The significant reduction in nutrient substances could result from the decrease in digestion and absorption and an increase in energy consumption of *B. odoriphaga*. The decrease in digestive enzyme activity and increase in antioxidant enzyme activity in the present study supported this possibility (Figure 2 and Figure 3). These data suggest that most of the ingested food is being converted to energy for combatting the invading pathogen and little food is being channeled toward larval growth. In addition, the production of toxic substances by the entomopathogenic fungus can disrupt insect food digestion and absorption [38]. A few entomopathogenic fungi have been reported to invade the insect body through the digestive system, where they disrupt larval feeding at the beginning of infection.

Entomopathogenic fungi can bypass the host defense system by several mechanisms. Entomopathogens produce toxins that stimulate oxidative mechanisms in insects, and this suggested that pathogens disable antioxidative enzymes to avoid the protective responses of insects to oxidative stress [15,39]. Karthi et al. (2018) reported that the infection by *Aspergillus flavus*, an entomopathogenic fungus, resulted in a significant increase in antioxidant enzyme activity (CAT, POD, and SOD) but a significant decline in phenoloxidase activity [17]. Our results indicated that *M. hiemalis* BO-1 infection triggers the POD enzyme level of *B. odoriphaga.* The SOD, CAT, and GST enzymes increased significantly at 12 to 24 h after infection (Figure 3), which indicated the importance of these enzymes in the defense against the oxidative stress caused by entomopathogenic fungi infection. However, there was a significant decline in SOD, CAT, and GST enzyme activities at 48 h after infection, and during this time, the infected larvae began to die (Figure 3). This indicated a decrease in the defense ability of insects, which could be attributed to the damage to physiological functions caused by entomopathogenic fungi [40]. A high concentration of fungal inoculum produced more significant effects (Figure 3). Similarly, Qasim et al. (2021) reported that SOD, POD, and GST enzymatic activity of *Diaphorina citri* Kuwayama adult and nymphal populations decreased significantly at 5 d posttreatment (fungal exposure) [20]. We conclude that entomopathogenic fungi had various effects on the antioxidant enzymes responses of insects such as *Aedes aegypti* [41], *Leptinotarsa decemlineata* [42], *Chilo suppressalis* [43], and *Periplaneta americana* [44]. Moreover, at 96 h after infection, a striking decrease in antioxidant enzyme activities (SOD and GST), digestive enzyme activities (protease, α-amylase, lipase, and cellulase), and nutrient substances (proteins, lipids, and sugars) was observed. This phenomenon may result from the severe biological stress caused by *M. hiemalis* infection, which is consistent with the pathogenicity results that the survival rate of infected larvae decreased dramatically from 72 h to 120 h after inoculation. In addition to the above results on digestive enzymes, our present study suggests that *M. hiemalis* BO-1 produces mycotoxins that disturb the physiology of *B. odoriphaga*.

Fungal infection is a complex process involving many factors as well as the activation of immune responses in the host insects [45]. To gain insight into the interactions between the insect host and the fungus, we explored the transcriptional response of *B. odoriphaga* larvae after infection with *M. hiemalis* BO-1, using RNA-seq technology. We identified 377 DEGs in infected *B. odoriphaga* larvae. Of these, 214 DEGs were up-regulated, and 163 DEGs were down-regulated. Transcriptional responses of *B. odoriphaga* to various conditions have been reported. Compared to a previous study [46], fewer DEGs were identified in the non-parametric transcriptome analysis of *B. odoriphaga* responses to insecticide stress (benzothiazole) in our study, and the quantity of DEGs identified in the transcriptome analysis using *B. odoriphaga* genome data was similar to that in previous studies [47,48]. Therefore, our transcriptome analysis based on the *B. odoriphaga* genome database should be more precise. Gene ontology (GO) analysis revealed that DEGs were especially enriched in terms associated with BP and CC. The 10 enriched biochemical pathways of the DEGs according to the KEGG database comprised the PPAR signaling pathway, metabolic pathways, lysosome, fatty acid metabolism, phagosome, ferroptosis, glycerolipid metabolism, fatty acid degradation, metabolism by cytochrome P450 and steroid biosynthesis. The RT-PCR results of the key targeted genes were consistent with the transcriptome results, for example, up-regulation of antioxidant genes (*PXDN*, *gst*, and *SOD1*), immune-related genes (*CYP9* and *ECSIT*) and energy metabolism genes (*treA* and *CPT1A*), and down-regulation of digestion and absorption-related genes (*uidA*, *amyA*, *PRSS*, and *LIPA*) (Figure 5 and Table 3). Our transcriptome analysis and physiological assay results confirmed that *M. hiemalis* BO-1 infection disturbed the physiological function of *B. odoriphaga* larvae, especially for energy metabolism, digestive absorption, and immune responses. Previous studies also reported that entomopathogenic fungi caused various effects on the metabolic and immune processes of insects such as *Bombyx mori* infected by *Beauveria bassiana* and *Helicoverpa zea* [49,50], *Diaphorina citri* infected by *Cordyceps fumosorosea* [51], and *Aedes albopictus* and *Culex pipiens* infected by *Pythium guiyangense* [52]. Interestingly, *M. hiemalis* BO-1 infection caused significant effects on the digestion and absorption pathway of *B. odoriphaga* larvae, which was possibly due to the unique pathogenic mechanism of *M. hiemalis* BO-1, such as causing infection via the enteric canal, damaging the intestinal structure, and then disturbing ingestion. The entomopathogenic bacteria *Bacillus thuringiensis* Aizawai GC-91 damaged the midgut cells of *Diatraea saccharalis* larvae [53]. The internal tissues of *Culex quinquefasciatus* larvae (midgut wall, skeletal muscles, and the cuticle-secreting epidermis) were destroyed by the entomopathogenic fungi, *Aspergillus clavatus* [54]. *Pythium guiyangense*, an oomycete entomopathogenic fungus, uses cuticle penetration and ingestion of mycelia into the digestive system to infect *Culex pipiens pallens* larvae [52]. Some studies also reported that the gut bacterial community of host insects can be altered by entomopathogenic fungi, and this accelerates larval mortality. Examples include *Dendroctonus valens* and *Beauveria bassiana* [55], *Anopheles stephensi* and *Beauveria bassiana* [56]. The disease symptoms, the results of food consumption, nutrient content and digestive enzyme activity suggest that *M. hiemalis* BO-1 may have similar infective functions. However, the interaction between entomological fungi and host insects changes with the infection process. The immune process of the insect is enhanced during the early stages of infection. However, as the disease progresses, the insect’s physiology is disrupted, and its immune processes become ineffective [50,57]. To understand the detailed interaction between *M. hiemalis* BO-1 and *B. odoriphaga*, the temporospatial modulation analysis of gene regulatory pathways should be further investigated.

## 5. Conclusions

This study showed that *M. hiemalis* BO-1 was highly pathogenic to *B. odoriphaga* larvae. Infected larvae died quickly, and the lethal efficiency was higher than other entomopathogenic fungi. *M. hiemalis* BO-1 infection resulted in decreased food consumption and digestive enzyme activity, disturbed energy metabolism, and material accumulation. Infection was also accompanied by fluctuations in immune function. The results suggested that infection of the intestinal canal and damage to the digestive function are important infection and pathogenic strategies of *M. hiemalis* BO-1.

## Figures and Tables

**Figure 1 insects-14-00162-f001:**
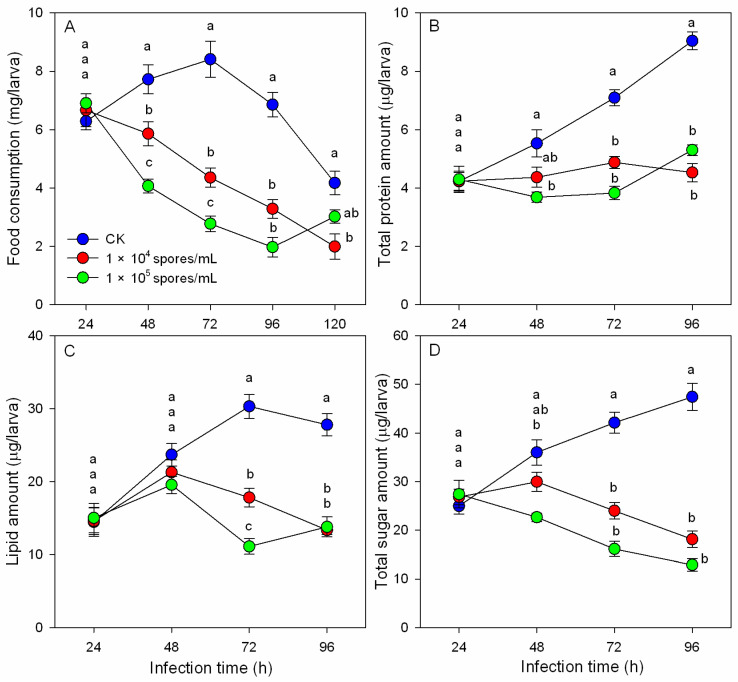
Effects of *M. hiemalis* infection on food consumption (**A**) and nutrient contents ((**B**), protein amount; (**C**), lipid amount; (**D**), sugar amount) of *B. odoriphaga.* Data in the figure are the mean ± SE. Different letters over the same column indicate significant differences between different spore concentration treatments at the *p* < 0.05 level by one-way ANOVA. Different letters (a and b) over the same column indicate significant differences between different spore concentration treatments at the *p* < 0.05 level as indicated by one-way ANOVA.

**Figure 2 insects-14-00162-f002:**
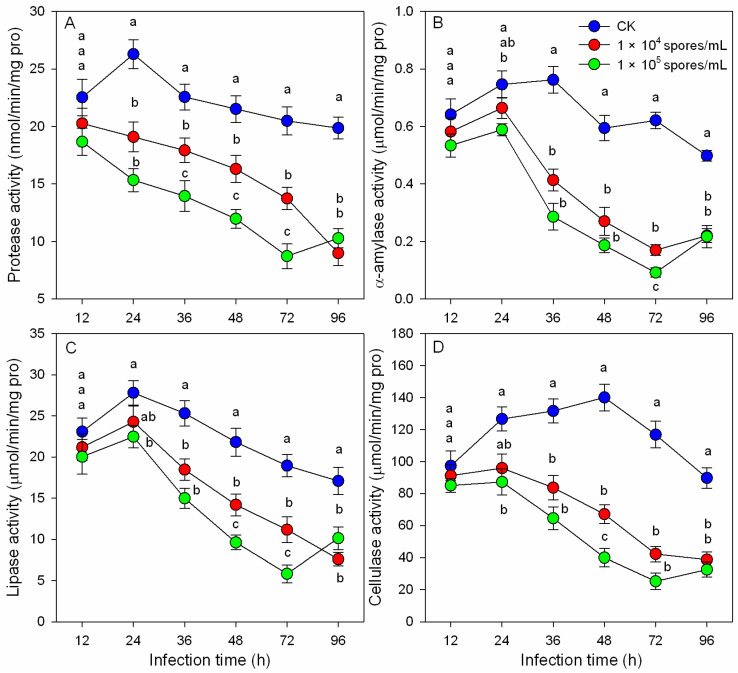
Digestive enzyme activities ((**A**): protease, (**B**): α-amylase, (**C**): lipase, and (**D**): cellulase) of *B. odoriphaga* larvae after infection with *M. hiemalis* spores. Data in the figure are the mean ± SE. Different letters over the same column indicate significant differences between different spore concentration treatments at the *p* < 0.05 level by one-way ANOVA. Different letters (a and b) over the same column indicate significant differences between different spore concentration treatments at the *p* < 0.05 level as indicated by one-way ANOVA.

**Figure 3 insects-14-00162-f003:**
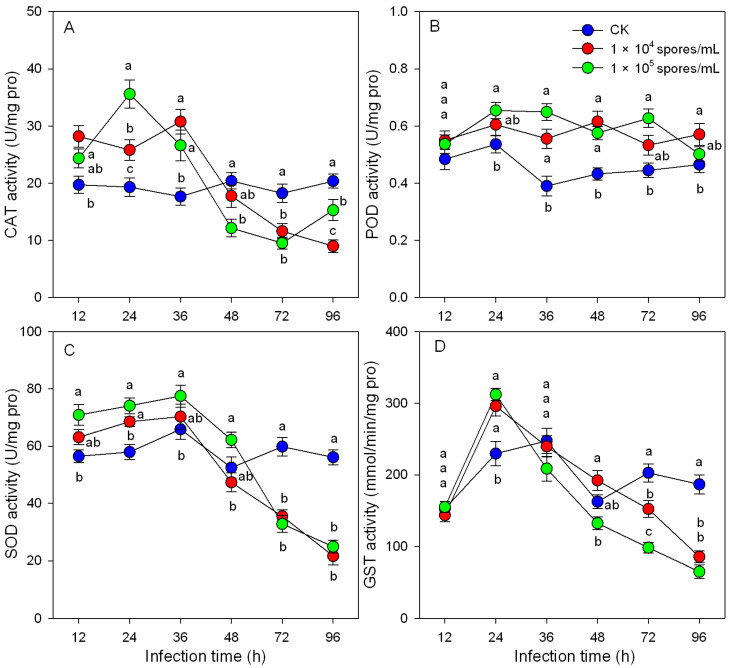
Antioxidant enzyme activities ((**A**): CAT, (**B**): POD, (**C**): SOD, and (**D**): GST) of *B. odoriphaga* larvae after infection with *M. hiemalis* spores. Data in the figure are the mean ± SE. Different letters over the same column indicate significant differences between different spore concentration treatments at the *p* < 0.05 level by one-way ANOVA.

**Figure 4 insects-14-00162-f004:**
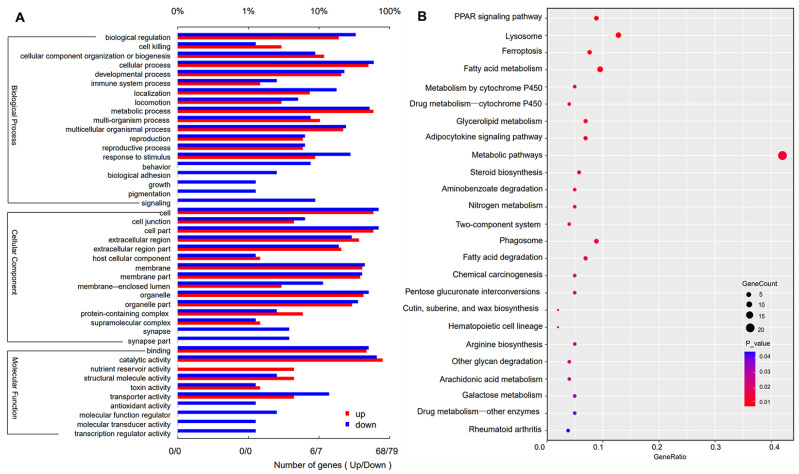
GO term enrichment analysis for the DEGs and the distribution of pathways annotated in the Kyoto Encyclopedia of Genes and Genomes (KEGG) data library in *B. odoriphaga* larvae after infection with *M. hiemalis* spores.

**Figure 5 insects-14-00162-f005:**
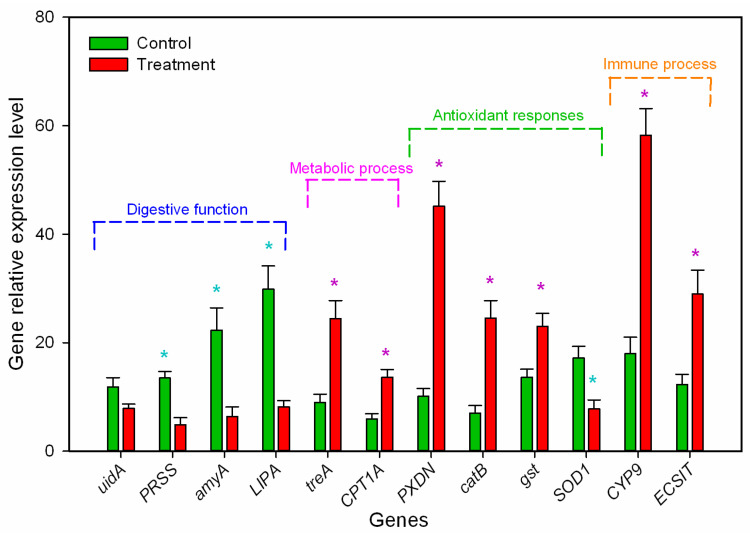
Real-time polymerase chain reaction (RT-PCR) analysis of gene expression in *B. odoriphaga* larvae after infection with *M. hiemalis* spores. RT-PCR data calculated with the 2^−∆∆Ct^ method. The value of the relative expression level on the y-axis was calculated according to log(2^−∆∆Ct^). *, *p* < 0.05, as determined by one-way analysis of variance (ANOVA).

**Table 1 insects-14-00162-t001:** Pathogenicity of *M. hiemalis* BO-1 against *B. odoriphaga* 4th-instar larvae.

**Infection Time** **(d)**	**Regression Equation** **(y = ax + b)**	**LC_50_ Median Lethal Concentration (spores/mL)**	**95% Confidence Interval** **(spores/mL)**	**R^2^**
3	y = −10.802x + 8.663	2.180 × 10^6^	1.099~4.736 × 10^6^	0.892
4	y = −9.888x + 7.143	1.891 × 10^5^	1.001~3.836 × 10^5^	0.983
5	y = −11.187x + 7.804	0.965 × 10^5^	0.551~1.729 × 10^5^	0.940
6	y = −7.384x + 4.798	2.914 × 10^4^	1.438~6.742 × 10^4^	0.999
**Spore concentration** **(spores/mL)**	**Regression equation** **(y = ax + b)**	**LT_50_ Median lethal time (d)**	**95% Confidence interval** **(d)**	**R^2^**
1.00 × 10^4^	y = −7.149x + 4.661	4.787	4.476~5.146	0.995
1.00 × 10^5^	y = −7.004x + 8.663	3.791	3.518~4.085	0.988
1.00 × 10^6^	y = −7.672x + 3.845	2.986	2.753~3.207	0.887
1.00 × 10^7^	y = −8.143x + 3.605	2.618	2.407~2.816	0.938
1.00 × 10^8^	y = −7.796x + 3.209	2.432	2.223~2.627	0.898

**Table 2 insects-14-00162-t002:** Aligning statistics of clean reads with assembled unigenes.

Samples	Total Reads	Clean Reads	Q30(%)	GC Content (%)	Mapped Reads	Mapped Ratio (%)
Control-1	7,485,207,667	50,812,192	94.22	41.92	45,067,128	88.69
Control-2	7,558,357,480	51,677,542	94.33	42.02	46,192,515	89.39
Treatment-1	7,141,260,179	49,670,662	94.19	42.21	43,655,545	87.89
Treatment-2	6,615,063,328	44,925,166	94.27	42.01	39,841,706	88.68

**Table 3 insects-14-00162-t003:** Differently expressed genes of *B. odoriphaga* larvae infected by *M. hiemalis* BO-1 by transcriptome analysis.

Gene ID	Gene Name	GO Biological Process	GO Function	KEGG Pathways	log2 Ratio (T/C)	*p*-Value
HA402_000028	*uidA*	GO0005975: Carbohydrate metabolic process	Beta-glucosidase activity	ko04973: Carbohydrate digestion and absorption	−2.05	3.96 × 10^-4^
HA402_001933	*PRSS*	GO0007586: Digestion	Serine-type endopeptidase activity	ko04974: Protein digestion and absorption	−1.85	8.96 × 10^-6^
HA402_001429	*amyA*	GO0005975: Carbohydrate metabolic process	Alpha-amylase activity	ko04973: Carbohydrate digestion and absorption	−2.40	5.38 × 10^-6^
HA402_001695	*LIPA*	GO0016042: Lipid digestion process	Lipase activity	ko00100: Steroid biosynthesis	−3.08	1.53 × 10^-6^
HA402_003401	*treA*	GO0005991: Trehalose metabolic process	Trehalase activity	ko01100: Metabolic pathways	1.96	2.97 × 10^-13^
HA402_014097	*CPT1A*	GO0006635: Fatty acid beta-oxidation	Palmitoleoyltransferase activity	ko01212: Fatty acid metabolism	2.54	2.97 × 10^-13^
HA402_012831	*PRDX*	GO0006979: Response to oxidative stress	Peroxidase activity	ko04146: Peroxisome	1.56	4.69 × 10^-4^
HA402_001366	*catB*	GO0042744: Hydrogen peroxide catabolic process	Catalase activity	ko04146: Peroxisome	2.31	4.24 × 10^-3^
HA402_005674	*gst*	GO0006979: Response to oxidative stress	Glutathione transferase activity	ko00480: Glutathione metabolism	−2.32	2.18 × 10^-7^
HA402_011746	*SOD1*	GO0019430: Removal of superoxide radicals	Superoxide dismutase activity	ko04146: Peroxisome	−1.59	1.11 × 10^-5^
HA402_008470	*CYP9*	GO0017144: Drug metabolic process	cytochrome P450 activity	ko00982: Drug metabolism—cytochrome P450	2.67	4.25 × 10^-6^
HA402_001791	*ECSIT*	GO0045087: Innate immune response	Hydrolase activity	ko04624: Toll and Imd signaling pathway	1.27	5.32 × 10^-4^

## Data Availability

The data about biological and physiological indexes are available at https://doi.org/10.5061/dryad.z34tmpgjg accessed on 16 November 2022, and the RNA-seq data are available at NCBI SRA, accession number PRJNA922622.

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
