# Peer review of "Effects of the Entomopathogenic Fungus Mucor hiemalis BO-1 on the Physical Functions and Transcriptional Signatures of Bradysia odoriphaga Larvae"

_insects, 2023, doi:10.3390/insects14020162_

Round 1
Reviewer 1 Report
Authors present the research on Bradysia odoriphaga larvae infected by Mucor hiemalis BO-1. The experimental design and methodology used in the manuscript seemed sound, but reviewer cannot access the URL indicated in ‘Data Availability Statement’ (https://doi.org/10.5061/dryad.37pvmcvm ). It is very critical to assess the manuscript. The supplemental materials must be opened in the review process, thus the manuscript cannot be assessed.
Therefore reviewer commented the manuscript based on the information in the main text.
First of all, authors reported the transcriptome analysis of B. odoriphaga, but any sequence data for that seemed not to be submitted to public repository. These data must be submitted to the International Nucleotide Sequence Databases, and in this case Sequence Read Archive (SRA; https://www.ncbi.nlm.nih.gov/sra/) is an appropriate repository. The manuscript must be reported with accession numbers for that database.
Furthermore, the analysis of DEGs in Figure 4 is informative and impressive, but it needs to be analyzed in more detail. What pathways in metabolic pathway were enriched? It must be reported with the list of genes in the supplemental materials author did not open in the current review.
Author Response
Comment 1: Authors present the research on Bradysia odoriphaga larvae infected by Mucor hiemalis BO-1. The experimental design and methodology used in the manuscript seemed sound, but reviewer cannot access the URL indicated in ‘Data Availability Statement’ (https://doi.org/10.5061/dryad.37pvmcvm). It is very critical to assess the manuscript. The supplemental materials must be opened in the review process, thus the manuscript cannot be assessed. Therefore, reviewer commented the manuscript based on the information in the main text.
Response: Thank you for this suggestion. We have resubmitted the data to the Dryad Data Platform, and paid for the data disclosure on January 12th, 2023. Now, the data can be accessed by the connection (Data Availability Statement, https://doi.org/10.5061/dryad.z34tmpgjg). We also mentioned this in the revised manuscript (Line 538).
Comment 2: First of all, authors reported the transcriptome analysis of B. odoriphaga, but any sequence data for that seemed not to be submitted to public repository. These data must be submitted to the International Nucleotide Sequence Databases, and in this case Sequence Read Archive (SRA; https://www.ncbi.nlm.nih.gov/sra/) is an appropriate repository. The manuscript must be reported with accession numbers for that database.
Response: Thank you for your suggestion. We have submitted the transcriptome data to the International Nucleotide Sequence Databases (Sequence Read Archive, SRA; https://www.ncbi.nlm.nih.gov/sra/), and the accession number is PRJNA612767 which mentioned in the revised manuscript (Line 240-241).
Comment 3: Furthermore, the analysis of DEGs in Figure 4 is informative and impressive, but it needs to be analyzed in more detail. What pathways in metabolic pathway were enriched? It must be reported with the list of genes in the supplemental materials author did not open in the current review.
Response: Thank you for this suggestion. We added some detailed analysis in the revised manuscript (line 363-line 373). “In the metabolic pathways, 23 DEGs were identified (Table S5). Genes including lipid catabolic processes (HA402-011554, HA402-013349), carbohydrate metabolic processes (HA402-007273, HA402-007340, HA402-005954), glutathione metabolic processes (HA402-015641, HA402-005674), glucuronosyltransferase activity (HA402-015641, HA402-015641) or phospholipid metabolic processes (HA402-011554) were partially repressed. Genes including trehalose metabolic processes (HA402-003401, HA402-006219, HA402-013707), monosaccharide metabolic processes (HA402-009149), lipid biosynthetic processes (HA402-002025), dopamine metabolic processes (HA402-005954, HA402-001421) or doxorubicin metabolic processes (HA402-001272) were upregulated.” In addition, we also added some detail results in the Supplementary Materials (Line 529-535), such as Table S3. Differentially expressed genes identified in Bradysia odoriphaga infected by Mucor hiemalis BO-1; Table S4. Detailed information for the KEGG enrichment analysis of the DEGs in the Bradysia odoriphaga transcriptome; Table S5. Detail information in metabolic pathway enriched in Bradysia odoriphaga infected by Mucor hiemalis BO-1. We think the supplementary analysis is adequate for this paper.
Reviewer 2 Report
The manuscript “Effects of the entomopathogenic fungus Mucor hiemalis BO-1 2 on the physical functions and transcriptional signatures of 3 Bradysia odoriphaga larvae” is a study focused on revealing some physical and molecular consequences in Bradysia odoriphaga larvae exposed to the entomopathogenic fungus Mucor hiemalis BO-1. To this end, the authors show the results derived from pathogenicity assays in the target insect, in addition to evaluating how food consumption, nutrient content, enzyme activity, and gene expression (using bulk RNAseq) are affected in larvae exposed to the fungus versus controls. The work is well diagrammed and written, and the general evidence is of interest for the proposal of novel biological pest control agents.
Some comments and questions
*How do you quantify fungal spores to carry out tests on the target insect? Please, introduce this information in section 2.2
*Please, homogenize the way of writing the volumes. For example, in line 137 it is written "ml" but in line 139, "mL". Check all the text regarding how the volumes are written (including figures).
*Please indicate on line 173 if a commercial kit was used or include the corresponding reference (Bradford, M., 1976. A Rapid and Sensitive Method for the Quantitation of Microgram Quantities of Protein Utilizing the Principle of Protein-Dye Binding. Analytical Biochemistry, 72(1-2), pp. 248-254).
*How was the quality of the RNA isolated from the larvae determined? Mention, if available, the RIN values.
*Given that a bulk RNAseq strategy was used on larvae treated with the fungus and untreated as a control, how were the reads cleaned up with respect to mRNA derived from the fungus or from the larval microbiome?
*Justify why a treatment with 1.0 × 10^5 spore/mL was used for the evaluation tests on the treated larvae (results from section 3.2 onwards).
*In sections 3.3, 3.4, and 3.5, how do you explain the 96-hour results among the group exposed to 1.0 × 10^4 spores/mL or 1.0 × 10^5 spores/mL? The change in trend in some of the measurements made is striking.
*Were the RT-qPCR assays performed on the same RNA samples used for RNAseq or are they derived from independent assays? Please clarify.
* Reports of other studies that have performed RNAseq in Bradysia odoriphaga are found. It would be interesting if some of the results found in this work could be linked to the previous ones.
Author Response
The manuscript “Effects of the entomopathogenic fungus Mucor hiemalis BO-1 on the physical functions and transcriptional signatures of Bradysia odoriphaga larvae” is a study focused on revealing some physical and molecular consequences in Bradysia odoriphaga larvae exposed to the entomopathogenic fungus Mucor hiemalis BO-1. To this end, the authors show the results derived from pathogenicity assays in the target insect, in addition to evaluating how food consumption, nutrient content, enzyme activity, and gene expression (using bulk RNAseq) are affected in larvae exposed to the fungus versus controls. The work is well diagrammed and written, and the general evidence is of interest for the proposal of novel biological pest control agents.
Comment 1: How do you quantify fungal spores to carry out tests on the target insect? Please, introduce this information in section 2.2
Response: Thank you for your suggestion. We added the description about the method of preparing the spore suspension in the revised manuscript (line 135-line 141). “This strain was cultivated on potato dextrose agar (PDA) plate (Φ = 9 cm) in Petri dishes at an optimum temperature of 23°C. After 10 d, 50 ml 0.1% Tween 80 distilled water was added to the dish, and the surface of the colony was gently and repeatedly scraped with a petri scalpel. The suspension was filtered using three-layer filter paper, and the filtrate was gathered as a spore suspension. The spore suspension concentration was calculated through blood counting chamber analysis.”
Comment 2: Please, homogenize the way of writing the volumes. For example, in line 137 it is written "ml" but in line 139, "mL". Check all the text regarding how the volumes are written (including figures).
Response: Thank you for your suggestion. We have homogenized the writing the volumes.
Comment 3: Please indicate on line 173 if a commercial kit was used or include the corresponding reference (Bradford, M., 1976. A Rapid and Sensitive Method for the Quantitation of Microgram Quantities of Protein Utilizing the Principle of Protein-Dye Binding. Analytical Biochemistry, 72(1-2), pp. 248-254).
Response: Thank you for your suggestion, and we have added this reference in the reference list, and numbered as No. 32.
Comment 4: How was the quality of the RNA isolated from the larvae determined? Mention, if available, the RIN values.
Response: Thank you for your suggestion, and we have added the RNA quality detection method in the revised manuscript (Line 234-236). “The RNA quality was monitored on 1% agarose gel, and RNA purity was confirmed using a NanoPhotometer spectrophotometer (Table S1).” The result of RNA quality was mentioned in the supplementary materials (Table S1).
Comment 5: Given that a bulk RNAseq strategy was used on larvae treated with the fungus and untreated as a control, how were the reads cleaned up with respect to mRNA derived from the fungus or from the larval microbiome?
Response: Thank you for your suggestion. To clean up the mRNA derived from the fungus or other microbiota in larvae, clean reads were mapped to the B. odoriphaga genome (Bioproject ID: PRJNA612767) (mentioned in the revised manuscript Line 242-244).
Comment 6: Justify why a treatment with 1.0 × 10^5 spore/mL was used for the evaluation tests on the treated larvae (results from section 3.2 onwards).
Response: Thank you for your suggestion. We added the introduction in the revised manuscript Line 154-158. “According to the pathogenicity results, the survival rates of fourth-instar larvae at 24, 48, 72 and 96 h after inoculation were 98.75%, 90%, 78.75% and 63.75% for 104 spores/ml treatment and 97.5%, 86.25%, 68.75% and 50% for 105 spores/ml treatment, respectively. At 96 h, the survival rate of infected larvae was close to 50%, and so the two spore concentrations of 104 spores/ml and 105 spores/ml were used for the evaluation tests on the treated larvae.” We think the survival rate of the diseased B. odoriphaga infected by 104 spores/ml and 105 spores/ml treatment within 96 h after inoculation was about 50%, which was suitable for the following analysis.
Comment 7: In sections 3.3, 3.4, and 3.5, how do you explain the 96-hour results among the group exposed to 1.0 × 10^4 spores/mL or 1.0 × 10^5 spores/mL? The change in trend in some of the measurements made is striking.
Response: Thank you for your suggestion. We added the discussion in the revised manuscript Line 453-458. “Moreover, at 96 h after infection, a striking decrease in antioxidant enzyme activities (SOD and GST), digestive enzyme activities (protease, α-amylase, lipase and cellulase), and nutrient substances (proteins, lipids and sugars) was observed. This phenomenon may result from the severe biological stress caused by M. hiemalis infection, which is consistent with the pathogenicity results that the survival rate of infected larvae decreased dramatically from 72 h to 120 h after inoculation.”
Comment 8: Were the RT-qPCR assays performed on the same RNA samples used for RNAseq or are they derived from independent assays? Please clarify.
Response: Thank you for your suggestion. We added the description in the revised manuscript Line 258-261. “To validate the results of the transcriptome analysis, 12 DEGs were analyzed using qRT-PCR. New biological samples were prepared according to the methods described in Section 2.4. Total RNA isolation and cDNA synthesis were performed”. So, this is a new qRT-PCR analysis compared with the above transcriptome analysis.
Comment 9: Reports of other studies that have performed RNAseq in Bradysia odoriphaga are found. It would be interesting if some of the results found in this work could be linked to the previous ones.
Response: Thank you for your suggestion. We added the discussion in the revised manuscript Line 466-471. “Transcriptional responses of B. odoriphaga to various conditions have been reported. Compared to a previous study[47], fewer DEGs were identified in the non-parametric transcriptome analysis of B. odoriphaga responses to insecticide stress (benzothiazole) in our study, and the quantity of DEGs identified in the transcriptome analysis using B. odoriphaga genome data was similar to that in previous studies[48,49]. Therefore, our transcriptome analysis based on the B. odoriphaga genome database should be more precise.” However, we did not find any reports about the transcriptome analysis of B. odoriphaga response to entomopathogen, which restricted the depth of our discussion.
Reviewer 3 Report
The paper can be accepted without any further changes.
M. hiemalis BO-1 infection resulted in decreased food consumption and digestive enzyme activity, disturbed energy metabolism and material accumulation. Infection was also accompanied by fluctuations in immune function. The results suggested that infection of the intestinal canal and damage to the digestive function are important infection and pathogenic strategies of M. hiemalis BO-1.
The key genes related pathogenic processes of Mucor hiemalis BO-1 infection still remained unknown,need more work to identify. We noticed the isolates optimal growth temperature was lower (23 C) of Mucor hiemalis BO-1, which might delay the infection processes; please explain why choose this type isolates instead of one with higher optimal temperature.
Author Response
Comment 1: We noticed the isolates optimal growth temperature was lower (23 C) of Mucor hiemalis BO-1, which might delay the infection processes; please explain why choose this type isolates instead of one with higher optimal temperature.
Response: Thank you for your suggestion, and we think your suggestion is helpful for our future research. In addition, our previous study reported that M. hiemalis BO-1 also possessed stronger pathogenicity against B. odoriphaga larvae at 18–28°C, and 23°C was the optimum temperature for growth, sporulation and pathogenicity of M. hiemalis BO-1 (Zhu, G., Ding, W., Xue, M., Zhao, Y., Li, M., Li, Z. Identification and pathogenicity of a new entomopathogenic fungus, Mucor hiemalis (Mucorales: Mucorales), on the root maggot, Bradysia odoriphaga (Diptera: Sciaridae). J. Insect Sci. 2022, 22(2), 1-9. https://doi.org/10.1093/jisesa/ieac010). Previous studies reported that during the emergence peak of B. odoriphaga larvae, the mean soil temperature fluctuated fluctuates between 15 and 25°C, which would be suitable for the use of M. hiemalis BO-1 to control of B. odoriphaga based on our results (Shi et al., 2018; Shi et al., 2020). And, compared with other entomopathogenic fungi of B. odoriphaga larvae, the application of M. hiemalis BO-1 is more promising.
Round 2
Reviewer 1 Report
Reviewer confirmed that data in the dryad repository (https://doi.org/10.5061/dryad.z34tmpgjg) can now be accessible properly.
Sequence data in SRA database associated with BioProject ID PRJNA612767 are all sequence from PacBio Sequel (ref: https://www.ncbi.nlm.nih.gov/sra?linkname=bioproject_sra_all&from_uid=612767). The information only with PRJNA612767 is not sufficient as reviewer cannot find corresponding transcriptome reads from that. Some data with the descriptions 'RNA_raw_data' and 'RNA_clean_data' seem to be generated from a short read sequencer, but the sequencer information (instrument) is 'Sequel'. If so, the descriptions in the database are not correct, and thus these must be fixed before the resubmission.
Author Response
Thank you very much for your careful review. I am very sorry to waste your valuable time because our negligence. We have corrected the database connection in the last revised manuscript (line 236), and the correct accession number is PRJNA922622. However, in the response letter we wrote the wrong accession number (PRJNA612767), which belong to the B. odoriphaga genome database reported by previous reports. We hope that our faults won't affect your approval of the manuscript. Thank you very much.
Round 3
Reviewer 1 Report
The data is now accurately described in the main text. However, the information about RNA-seq reads with BioProject ID (PRJNA922622) was not described in 'Data Availability Statement'. Add this information to this section for the reuse of data produced in this work.
Author Response
Thank you very much for your careful review. We have added the the information about RNA-seq reads with BioProject ID (PRJNA922622) in 'Data Availability Statement'(line 536-537:the RNA-seq data are available at NCBI SRA, accession number PRJNA922622).